# Self-Supervision is All You Need for Solving Rubik's Cube

**Kyo Takano**                                                                                     *kyo.takano@mentalese.co*

**Reviewed on OpenReview:** *https://openreview.net/forum?id=bnBeNFB27b*

## Abstract

Existing combinatorial search methods are often complex and require some level of expertise. This work introduces a simple and efficient deep learning method for solving combinatorial problems with a predefined goal, represented by Rubik's Cube. We demonstrate that, for such problems, training a deep neural network on random scrambles branching from the goal state is sufficient to achieve near-optimal solutions. When tested on Rubik's Cube, 15 Puzzle, and 7×7 Lights Out, our method outperformed the previous state-of-the-art method DeepCubeA, improving the trade-off between solution optimality and computational cost, despite significantly less training data. Furthermore, we investigate the scaling law of our Rubik's Cube solver with respect to model size and training data volume. Our code is available at github.com/kyo-takano/efficientcube

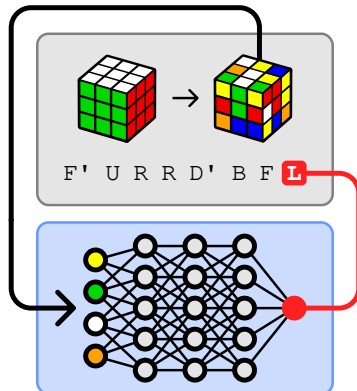

Figure 1: The proposed learning algorithm for solving Rubik's Cube. We train a Deep Neural Network (DNN) on sequences of random moves applied to the predefined goal. At each step, the DNN learns to predict the last scramble move based on the resultant state. To solve a scrambled state, we sequentially apply the reverse of the moves predicted by the trained DNN.

## 1 Introduction

Combinatorial search is important both in theory and practice. One representative example is *Traveling Salesman Problem*, in which a salesman tries to visit multiple cities in the shortest path possible (Applegate et al., 2011; Russell & Norvig, 2021). Despite how easy it may sound, the problem is classified as *NP-hard* due to its combinatorial complexity (Papadimitriou & Steiglitz, 1998); as the number of cities grows, the number of possible combinations to search for explodes, easily exceeding the limits of a classical computer. In the real world, algorithms for such problems have many applications, including planning, scheduling, resource allocation, and so on.

Moving forward from the impracticality of a naive brute-force search, a variety of search methods have emerged over time, aiming to address the computational complexity of combinatorial problems. These

include exact algorithms to systematically find an optimal solution within an acceptable time frame, and heuristic algorithms to quickly find a reasonably good solution. In recent years, deep learning has established itself as a potent method for automatically training a heuristic function optimized for a given objective. Deep reinforcement learning, in particular, has shown promise for training a Deep Neural Network (DNN) to solve a combinatorial problem near-optimally, even without labeled data or extensive memory (Mazyavkina et al., 2021; Vesselinova et al., 2020). By rewarding efficient moves/actions, a DNN can learn to find increasingly better solutions.

Nevertheless, combinatorial search remains a challenging task. Implementing an explicit search algorithm often necessitates domain expertise, substantial computational resources, and extensive human effort. Reinforcement learning, though it can automate parts of the process, still demands a comprehensive understanding of the methodology itself and significant amount of time spent in trial and error.

In the present study, we propose a novel method to solve a specific class of combinatorial problems—namely, those with a predefined goal. The idea is straightforward: train a DNN on mere random scrambles branching from the goal, as illustrated in Figure 1. We evaluate our method in comparison to DeepCubeA (Agostinelli et al., 2019), the previous state-of-the-art deep learning method, by experimenting with *Rubik's Cube* and a few other related problems.

## 2   Related Work

Rubik's Cube is a quintessential example of goal-predefined combinatorial problems. This three-dimensional puzzle consists of six faces, each with nine color stickers, and the objective is to manipulate the faces so that each face displays only a single color. With approximately $4.3 \times 10^{19}$ unique states, optimally solving this puzzle is an *NP-complete* problem(Demaine et al., 2018). To generalize, the task in problems like Rubik's Cube is to find a sequence of moves that will reach the predefined goal, starting from a given state. Henceforth, we view such combinatorial problems as a pathfinding task on an *implicit* graph, where each node represents a unique state and edges indicate moves connecting adjacent nodes.

Among various methods for solving these problems, we pay particular attention to two leading methods known for their high levels of optimality: Iterative Deepening A* (IDA*) and DeepCubeA. IDA* is a complete search algorithm that is guaranteed to find an optimal solution in a reasonable time frame (Korf, 1985; 1997). DeepCubeA, on the other hand, is a near-optimal method that leverages deep reinforcement learning (Agostinelli et al., 2019). Both are forward search methods to find a solution path by expanding nodes from a given state, in order of their estimated distances.

Korf (1985) invented IDA* as the first admissible search algorithm to optimally solve 15 Puzzle, another goal-predefined combinatorial problem (see Section 6.1 for problem description). It merges two algorithms: **I**terative **D**eepening Depth-First Search (IDDFS) and **A\*** search (Hart et al., 1968). IDDFS is a depth-first search that iteratively deepens until a solution is found. Like the breadth-first search, it is a complete search algorithm, but it can be infeasible to run within a reasonable time frame for large state spaces. To enable IDDFS in such vast state spaces, Korf (1985) incorporated the idea of A* search, which expands nodes in order of their estimated total distance formulated as

$$f(x) = g(x) + h(x) \tag{1}$$

where $x$ is an expanded node, $g(x)$ is the depth from the starting node, and $h(x)$ is the *lower bound* of the remaining distance to the goal informed by group theory. Using the same formula, IDA* iteratively increases the threshold for $f(x)$ instead of depth, greatly reducing the number of expanded nodes without compromising the optimality. Later, Korf (1997) extended IDA* as the first optimal Rubik's Cube solver, incorporating a pattern database that precomputes lower bounds of distances corresponding to particular patterns. Since then, more efficient implementations followed this approach (Korf, 2008; Arfaee et al., 2011; Rokicki et al., 2014). Even so, the implementation of IDA* remains problem-specific and requires domain knowledge of group theory and search.

DeepCube was the first to formulate Rubik's Cube as a reinforcement learning task and successfully solve it (McAleer et al., 2018), which soon evolved as DeepCubeA with significant performance improvement (Agostinelli et al., 2019). DeepCubeA trains a DNN in lieu of $h(x)$ in Equation (1) to estimate the distance from a given state to the goal. By relatively discounting $g(x)$ like weighted A* (Pohl, 1970; Ebendt & Drechsler, 2009) and expanding a certain number of nodes per iteration, DeepCubeA then searches for a solution so that the approximate lower bound of total distance $f(x)$ is as small as possible. DeepCubeA is also shown applicable to similar problems like 15 Puzzle and its variants, as well as Lights Out and Sokoban (Agostinelli et al., 2019).

In contrast to IDA*, which relies on group theory to estimate distance, DeepCubeA automatically trains a DNN to serve as a distance approximator. The use of DNN is well-suited for this task because it effectively learns complex patterns in data with a fixed number of parameters and also generalizes well to instances unseen during training. This adaptability is particularly convenient when a state space is too large to exhaustively explore and store in memory. However, it is essential to recognize that deep reinforcement learning carries inherent complexities and instability, necessitating meticulous design and hyperparameter tuning. In training DeepCubeA models, Agostinelli et al. (2019) had to check regularly that the training loss falls below a certain threshold to avoid local optima and performance degradation.

Although both are effective methods, they still require complex implementation with specialized knowledge. To this end, we propose a more direct method to infer the path itself, without the need for distance estimation to guide a search. Our method involves training a DNN on random scrambles from the goal, treating them as reverse solutions from the resulting states.

## 3 Proposed Method

We propose a novel self-supervised learning method for solving goal-predefined combinatorial problems, regarding the task as *unscrambling*—the act of tracing a scramble path *backward* to the goal. Although the actual scramble is not observable, we aim to statistically infer a plausible sequence of moves that would lead to the current state. At the heart of our method is a DNN trained to predict the last random move applied to a given state. We find a solution path by sequentially reversing the predicted moves from a scrambled state. Our method assumes and capitalizes on the inherent bias of random scrambles, which originate from the goal, toward optimality. The miniature example in Figure 2 illustrates the operations of our proposed method during training and inference.

When training a DNN, our method initializes the target problem with its goal state and applies a sequence of random moves to scramble it. At each step, the DNN learns to predict the last move applied based on the current state's pattern. As the training loss, we compute the categorical cross-entropy between actual and predicted probability distributions of the last move. Algorithm 1 outlines the training process, and Figure 1 presents an example data point on Rubik's Cube.

We search for a solution path tracing back to the goal state by sequentially reversing the DNN-predicted moves. We employ a best-first search algorithm and prioritize the most promising candidate paths, which we evaluate based on the cumulative product of the probabilities of all their constituent moves. AThe cumulative product can be represented as $\prod_{i=1} \hat{p}_i$, where $\hat{p}_i$ denotes the predicted probability that the $i$-th move is a reverse move toward the goal.

---

**Algorithm 1** The proposed training algorithm.

**Require:**
$G$: Goal state
$\mathbb{M}$: Set of moves
$B$: Number of scrambles per batch
$N$: Number of moves per scramble
$f_\theta$: DNN parameterized by $\theta$

**Ensure:** $f_\theta$: Trained DNN

**while** not converged **do**
  $T \leftarrow \varnothing$
  **for** $b = 1$ **to** $B$ **do**
    $S \leftarrow G$
    **for** $n = 1$ **to** $N$ **do**
      Sample m $\leftarrow$ random($\mathbb{M}$)
      Update $S \leftarrow S \circ$ m
      Add $(S, \text{m})$ to $T$
  Update $f_\theta$ using $T$

---

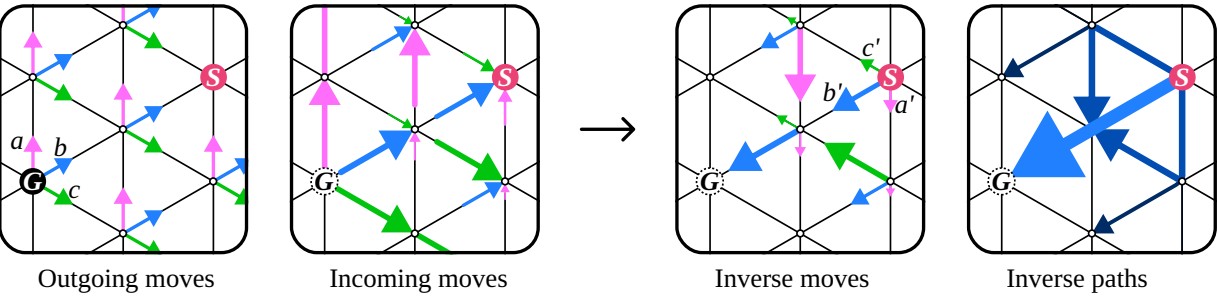

Figure 2: A miniature instance of combinatorial search with a predefined goal. Arrows symbolize moves, with their lengths and widths indicating the associated probabilities. *Left*: During training, random moves $\{a, b, c\}$ are feasible at each node with an equal probability of 1/3. These moves, in combination, form probabilistic paths from **G**oal to **S**cramble. By accumulating the probabilities of paths terminating with each move, we derive the probability distribution of incoming moves at every node. *Right*: During inference, we follow the learned distribution to traverse backward from **S** to **G** with the reverse moves $\{a', b', c'\}$. Candidate paths are evaluated and ranked based on the cumulative product of their estimated probabilities. In this instance, the path $[b', b']$ is estimated to be the most optimal among the other two-move paths.

Our method leverages a fundamental property of combinatorial search: the shorter a *path*, the more likely it is to occur randomly. This means the cumulative probability of a random training scramble increases as the number of moves decreases: $1/|\mathbb{M}|^N$, where $\mathbb{M}$ represents the set of moves and $N$ denotes the path length. Note that since different paths with various probabilities are convoluted for a specific state/pattern in the training process, the most frequent moves might not always comprise optimal paths. To mitigate the impact of such latent irregularities during inference, we evaluate candidate paths based on their cumulative probabilities, rather than merely the probability of the latest move. Based on these, we assume that generating random scrambles from the goal state establishes a correlation between the optimality of paths and their estimated probabilities.

Lastly, we note that God's Number—the minimum number of moves required to solve any state—or its upper bound should be known for the problem of interest. Let $N$ be the scramble length and $N_G$ be God's Number. If $N < N_G$, the trained DNN would not generalize well to more complex states requiring more than $N$ moves during inference. On the other hand, training data covers all possible complexities as long as $N_G \leqslant N$. Although redundant paths with more than $N_G$ moves would undermine the training efficiency, they have a negligible impact on the learned distribution due to the significantly lower probabilities than shorter ones. When neither $N_G$ nor its upper bound is known, it may be beneficial to gradually increase $N$ to the point where the DNN cannot make better-than-chance predictions. If the $N$-th moves are unpredictable, it implies that $N$ is exceeding the maximum complexity of the problem ($N > N_G$).

## 4 Experiment

We assess the effectiveness of our proposed method for solving Rubik's Cube, using the same problem setting, representation, dataset, and DNN architecture as DeepCubeA (Agostinelli et al., 2019). We then compare our result to the published result of DeepCubeA and its update provided on GitHub[1].

### 4.1 Rubik's Cube

We represent Rubik's Cube as a 324-D vector by assigning an index to each color at every sticker location (6 colors × 54 sticker positions). To manipulate states, we use the quarter-turn metric (a 90° turn of a face counts as one move, whereas a 180° turn counts as two), meaning 12 possible moves for any state. In this notation, U, D, L, R, B, and F denote a 90° clockwise turn on **U**p, **D**own, **L**eft, **R**ight, **B**ack, and **F**ront faces of the puzzle, respectively; if followed by a prime ('), counterclockwise.

---

[1]github.com/forestagostinelli/DeepCubeA/

### 4.2  Dataset

We utilize the publicly available DeepCubeA dataset on Code Ocean[2], which comprises $1,000$ Rubik's Cube test cases. Each case is scrambled with a range of $1,000$ to $10,000$ moves.

### 4.3  Model

The DNN consists of two linear layers (5000-D and 1000-D), followed by four residual blocks (He et al., 2016) each containing two 1000-D linear layers. Rectified Linear Unit (ReLU) activation (Nair & Hinton, 2010; Glorot et al., 2011) and batch normalization (Ioffe & Szegedy, 2015) are applied after each linear layer. Finally, the DNN has a 12-D linear layer to return logits as an output, which are then transformed into a probability distribution using the Softmax function.

### 4.4  Training

We train a DNN following the self-supervised approach outlined in Algorithm 1, where the DNN learns to predict the last move of random scrambles based on the resulting state. The DNN parameters are updated using Adam optimizer (Kingma & Ba, 2014) with an initial learning rate of $10^{-3}$. In this experiment, we set the maximum scramble length to 26, which is God's Number for Rubik's Cube in the quarter-turn metric (Kunkle & Cooperman, 2007). Additionally, the training process excludes clearly redundant or self-canceling permutations, such as R following R', when generating random scrambles. We train the DNN for $2,000,000$ steps with a batch size of $26,000$ (26 moves per scramble $\times 1,000$ scrambles per batch), which is equivalent to 2 billion solutions.

### 4.5  Inference

We use beam search, a heuristic search algorithm known for its efficiency, to solve Rubik's Cube using the trained DNN. While not guaranteed to always reach the goal, beam search can efficiently find solutions by prioritizing the most promising candidates.

Starting from depth 1 with a given scrambled state, at every depth $i$, the DNN predicts the probability distribution of the next possible moves for each candidate state. We set a beam width of $k$ and only pass the top $k$ candidate paths to the next depth $i + 1$, sorted by their cumulative product of probabilities. The search proceeds until any one of the candidate paths reaches the goal, at which point the search depth $i$ matches the solution length.

To investigate the trade-off between the number of nodes expanded and the optimality of solutions, we systematically increase the beam width $k$ from $2^0$ to $2^{18}$ in powers of 2. This also allows us to test whether our method needs to expand more nodes than DeepCubeA to achieve the same level of solution optimality.

## 5  Results

Figure 3 displays the performances of different methods based on two parameters: the solution length and the number of nodes expanded. For reference, we also include the performance of an optimal solver by Agostinelli et al. (2019), which extended Rokicki et al. (2014)'s implementation of IDA* that relies on 182 GB of memory. With $k \geqslant 2^7$, our method effectively solved all test cases using the trained DNN. The result also reveals a smooth trade-off between computational load and solution optimality, with a stably predictable number of nodes to expand for a specific beam width. Below, we specifically report our result obtained with the maximum beam width $k = 2^{18}$.

Alongside Figure 3, we summarize the comparative performance of the different methods in Table 1. We aligned the temporal performance of our method and DeepCubeA's paper result with the mean per-node time of DeepCubeA's GitHub result, in order to account for external factors affecting the metric, such as the number of GPUs and code quality [3].

---

[2]`doi.org/10.24433/CO.4958495.v1`
[3]We used a single GPU (NVIDIA T4), whereas Agostinelli et al. (2019) employed four GPUs (NVIDIA TITAN V).

Table 1: Performances of different methods in solving Rubik's Cube. We present average values for solution length, number of nodes, and time taken to solve per test scramble. Optimality (%) denotes the percentage of optimal solutions achieved by each method. For our method and DeepCubeA's paper result, the time taken is normalized based on the per-node temporal performance of DeepCubeA's GitHub result, with the actual wall-clock time in parenthesis for reference.

| Method | Solution length | Optimal (%) | Number of nodes | Time taken (s) |
|---|---|---|---|---|
| Optimal solver | 20.64 | 100.0 | $2.05 \times 10^6$ | 2.20 |
| **Ours** | **21.26** | **69.6** | $\mathbf{4.18 \times 10^6}$ | **38.73** (483.61) |
| DeepCubeA (GitHub) | 21.35 | 65.0 | $8.19 \times 10^6$ | 75.61 |
| DeepCubeA (Paper) | 21.50 | 60.3 | $6.62 \times 10^6$ | 61.14 (24.22) |

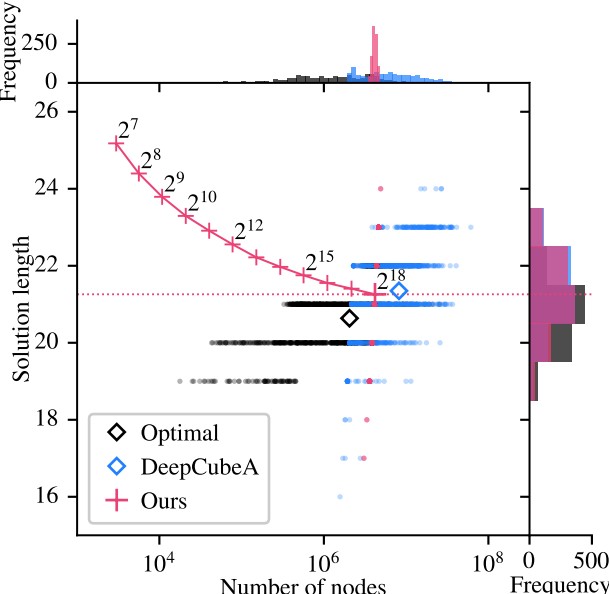

Figure 3: Joint distributions of solution length and the number of nodes expanded in solving Rubik's Cube by different methods. Each colored dot represents a unique solution, while diamond markers signify their mean coordinates by methods. Frequency distributions for each parameter are presented along each axis. The pink solid line indicates our method's trade-off between solution optimality and node count, annotated by the increasing beam widths.

The comparison indicates that our method outperforms DeepCubeA in terms of solution length and search efficiency, albeit not matching the optimal solver backed by 182 GB of memory. First, our solutions are more optimal than those of DeepCubeA on average. While optimal solutions average 20.64 moves in length, our method produced an average solution length of 21.26 moves, with 696 of the 1,000 solutions being optimal. On both metrics of optimality, our method surpassed that of DeepCubeA, whose solutions averaged 21.35 moves with a 65.0% optimality rate. Moreover, our solutions were also more computationally search-efficient. Whereas DeepCubeA expanded an average of $8.19 \times 10^6$ nodes per solution, ours expanded only about half the number of nodes to achieve the higher optimality.

Remarkably, our method achieved this superior result also with higher training efficiency. While DeepCubeA learned from 10 billion examples ($1,000,000$ iterations$\times 10,000$ scrambles per iteration), our method efficiently trained the DNN on the equivalent of only 2 billion examples. These findings underscore the overall advantage of our proposed method over DeepCubeA in solving Rubik's Cube.

# 6    Additional Evaluations

To see the versatility of this method, we extend our evaluation to two additional goal-predefined combinatorial problems: 15 Puzzle and 7×7 Lights Out. Like in the preceding Rubik's Cube Experiment, for both of these problems, we train a DNN of the same architecture and test it on the DeepCubeA dataset, which contains 500 cases each. While Agostinelli et al. (2019) also trained a DNN on 10 billion examples for each problem, we train ours with substantially less data to demonstrate the training efficiency of our method.

## 6.1    15 Puzzle

15 Puzzle is a 4×4 sliding puzzle consisting of 15 numbered tiles and one empty slot. The goal is to arrange these tiles in ascending numerical order by swapping the empty slot with its neighboring tiles multiple times. There are approximately $1.0 \times 10^{13}$ possible states for the puzzle, with God's Number standing at 80 moves (Brüngger et al., 1999; Korf, 2008).

We trained a DNN on the equivalent of 100 million solutions ($100,000$ batches $\times 1,000$ scrambles per batch), each scramble length being 80. This training data volume is equivalent to 1% of DeepCubeA, which used 10 billion training examples. Subsequently, we tested the trained DNN on 15 Puzzle subset of the DeepCubeA dataset. We note that, in similarity to the Rubik's Cube experiment, we removed redundant moves like [→] after [←], in both the training and inference phases.

Figure 4 and Table 2 show the comparative performances of different methods, including the IDA* optimal solver by Agostinelli et al. (2019) as a reference, which runs on 8.51 GB of memory. As illustrated in Table 2, our method solved all test cases using beam width $2^0 \leqslant k \leqslant 2^{16}$, progressively finding increasingly optimal solutions by expanding more nodes. With $k = 2^{16}$, we obtained optimal solutions for all test cases, while expanding 22.6% fewer nodes than DeepCubeA.

## 6.2    7×7 Lights Out

Lights Out is a combinatorial puzzle composed of multiple ON/OFF binary lights. In the 7×7 version, 49 lights are arranged in a square grid. Each light toggles the state of itself and its adjacent lights, and the goal is to switch off all the lights starting from a given random state. As a goal-predefined combinatorial problem, it possesses a more relaxed setting due to the following characteristics. First, because every move is binary, applying the same move twice is clearly redundant. Also, the puzzle is commutative, meaning that the order of moves applied is irrelevant to the resulting state. This characteristic suggests that each move within a given sequence holds an equal probability of being the last move. Hence, for a DNN, the problem is essentially a pattern recognition task to predict the *combination* of moves, rather than specifically the last move in the permutation. According to Agostinelli et al. (2019), an optimal solution is identified when the list of moves contains no repetitions.

Since God's Number is unknown for this particular problem, we set the training scramble length to 49 moves, the upper bound for when you need to toggle every button. We trained a DNN of the same architecture to predict *all* moves applied to the problem up to a given point in a scramble. In total, the DNN learned from the equivalent of 10 million solutions ($10,000$ batches $\times 1,000$ scrambles/batch).

As described in Table 3, our method solved all test cases optimally using a greedy search, which expands only the most prospective node at every depth. Despite using only 0.1% equivalent of training examples compared to DeepCubeA, our method significantly outperformed it in search efficiency, expanding only necessary nodes composing the optimal solutions. However, it must be noted that this flawless result on this particular dataset does not certify the completeness of the trained DNN as a search heuristic.

Table 2: Performances of different methods in solving 15 Puzzle. We present average values for solution length, number of nodes, and time taken to solve per test scramble. Optimality (%) denotes the percentage of optimal solutions achieved by each method. For our method and DeepCubeA's paper result, the time taken is normalized based on the per-node temporal performance of DeepCubeA's GitHub result, with the actual wall-clock time in parenthesis for reference.

| Method | Solution length | Optimal (%) | Number of nodes | Time taken (s) |
|---|---|---|---|---|
| Optimal solver | 52.02 | 100.0 | $3.22 \times 10^4$ | 0.0019 |
| **Ours** | **52.02** | **100.0** | $\mathbf{2.54 \times 10^6}$ | **6.83** (313.89) |
| DeepCubeA (GitHub) | **52.02** | **100.0** | $3.28 \times 10^6$ | 8.82 |
| DeepCubeA (Paper) | 52.03 | 99.4 | $3.85 \times 10^6$ | 10.37 (10.28) |

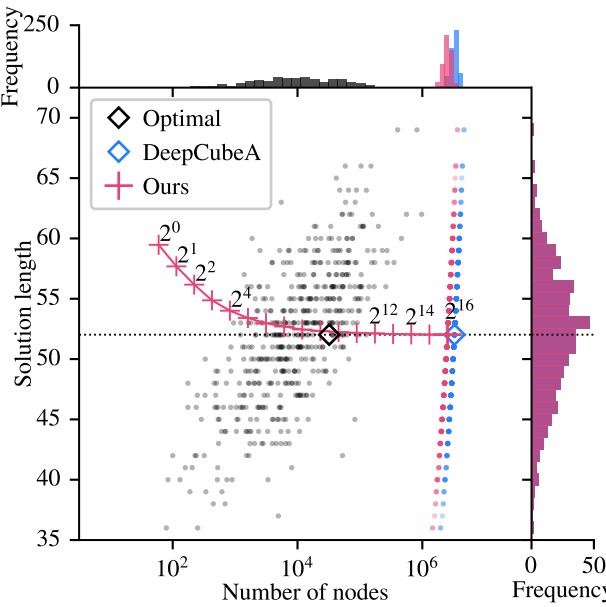

Figure 4: Joint distributions of solution length and the number of nodes expanded in solving 15 Puzzle by different methods. Each colored dot represents a unique solution, while diamond markers signify their mean coordinates by methods. Frequency distributions for each parameter are presented along each axis. The pink solid line indicates our method's trade-off between solution optimality and node count, annotated by the increasing beam widths.

Table 3: Performances of our method and DeepCubeA in solving 7×7 Lights Out. Since both our method and DeepCubeA reached optimal solutions across all cases, we report average values for solution length, the number of nodes, and the wall-clock time *without* normalization. We have omitted DeepCubeA's paper result here since both solution length and node count are consistent with its GitHub result.

| Method | Solution length | Number of nodes | Wall-clock time (s) |
|---|---|---|---|
| **Ours** | **24.26** | **24.26** | **0.053** |
| DeepCubeA | **24.26** | $1.14 \times 10^6$ | 5.90 |

# 7   Scaling Law

A DNN can better approximate a target data distribution with an increased number of parameters ($N$) or training data samples ($D$), as demonstrated across multiple domains (Hestness et al., 2017; Rosenfeld et al., 2019; Kaplan et al., 2020; Hoffmann et al., 2022; Zhai et al., 2022). Since our method involves training a DNN too, we expect to see the same scalability in our Rubik's Cube solver. However, these two factors—model and data sizes—have a trade-off relationship for a fixed FLOPs budget ($C$). Thus, it is crucial to estimate the optimal balance between $N$ and $D$ as a function of a given compute budget $C$. In this section, we formulate such a *scaling law* for our neural Rubik's Cube solver to obtain the best possible model. We subsequently evaluate the scaled models on the downstream task of solving the puzzle.

To estimate the scaling law for our Rubik's Cube solver, we adopt Hoffmann et al. (2022)'s power-law formulation for large language models. Hoffmann et al. (2022) defined the optimal allocation of training compute as a task of minimizing the cross-entropy loss $L(N, D)$, where $C \approx 6ND$ for training Transformers. They explored three approaches, one of which fitted the following parametric function:

$$\hat{L}(N, D) \triangleq E + \frac{A}{N^\alpha} + \frac{B}{D^\beta} \tag{2}$$

Here, $E$, $A$, $B$, $\alpha$, and $\beta$ are parameters to be estimated by the L-BFGS algorithm (Nocedal, 1980) using a set of observations ($N$, $D$, $L$) Following this approach, we estimated the scaling law for our solver by minimizing the mean squared error between $\hat{L}(N, D)$ and $L$. To fit the parameters with L-BFGS, we used a grid of the following initial values: $A$ and $B$ from $\{1, 2, \ldots, 10\}$, and $\alpha$ and $\beta$ from $\{0.1, 0.2, \ldots, 0.5\}$. We note that $C \approx 3ND$ for our DNNs, which primarily consist of dense linear layers.

We initiated the experiment by training 30 random low-budget DNNs, each with a compute budget between $1.4 \times 10^{13}$ and $3.2 \times 10^{15}$. We subsequently started fitting Equation (2) and *progressively* increased the compute budget, updating the optimal $N$ and $D$ estimation at each scaling step to train the next model. This strategy helped us to *gradually* explore the scaling space and reduce uncertainty attributed to extrapolation in higher-compute regions. It should be mentioned that this experiment differs from the main experiment (Section 4) in a few aspects. First, every DNN has a uniform number of latent dimensions per layer, and there are no residual connections. We also employed the half-turn metric, whereby both 90° and 180° turns count as one and God's Number for the puzzle is 20 (Rokicki et al., 2014). Accordingly, we set the scramble length to 20.

Figure 5 depicts the estimated scaling law, model data points ($N, D, L$), as well as the trajectory of progressively scaling our Rubik's Cube solver. We fitted a total of 36 models and empirically estimated the parameters for Equation (2) as $E = 0.892$, $A = 4.0$, $B = 6.2$, $\alpha = 0.178$, and $\beta = 0.134$. The $\alpha : \beta$ ratio of $0.57 : 0.43$ suggests that model size and training data volume should be scaled in roughly equal proportions for optimal performance, aligning with Hoffmann et al. (2022)'s finding on autoregressive language models. The final model was trained with $C = 5.14 \times 10^{19}$ FLOPs, allocated to $N \approx 119$ million parameters and $D \approx 144$ billion states (7.2 billion examples). We used about $12.1\%$ of the total compute to estimate the scaling law while dedicating the remaining $87.9\%$ to train the largest model.

We further evaluated the three largest models' capabilities in solving Rubik's Cube, using the DeepCubeA dataset once again. To evaluate them not merely on the basis of solution length but also the optimality rate, we employed IDA* (Korf, 2008) to prepare optimal solutions in the half-turn metric. Figure 6 affirms that increased training compute resulted in improved performance, yielding shorter and more optimal solutions with fewer nodes expanded. This result supports our assumption that the probability distribution of random scrambles, which the DNNs approximate, correlates with solution optimality. On the other hand, the result also reveals the near-saturating benefit of scaling model size for temporal performance, plausibly due to the increased latency of running larger models. Since temporal efficiency is also an important aspect of combinatorial search, one could also consider estimating the scaling law for optimal *temporal* performance, although this would heavily depend on the execution environment. In our experimental setup[4], we would benefit from increasing training data, but not model parameters.

---

[4]We used a single NVIDIA T4 GPU

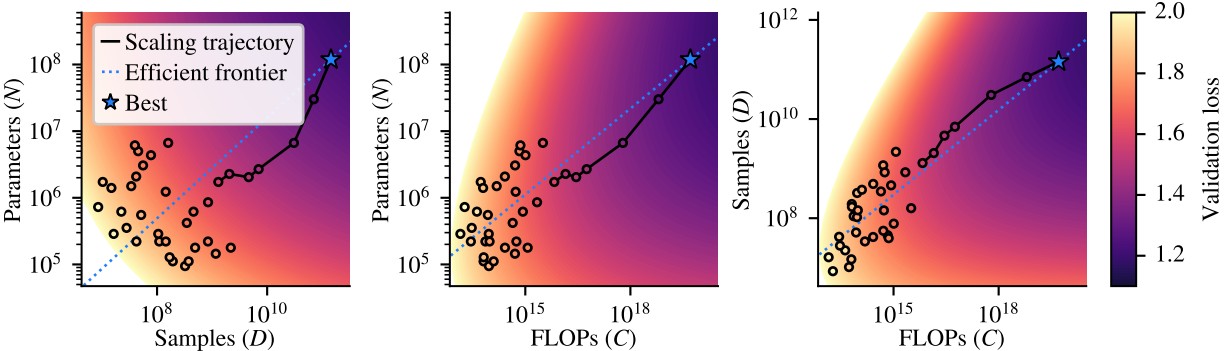

Figure 5: The scaling law estimation for our Rubik's Cube solver. The gradient surface in each subplot represents Equation (2) with the empirically estimated parameters. The blue dashed line indicates the compute-optimal frontier, i.e., the theoretical optimum to train the best model for a given compute budget. Each colored dot represents a unique model as a data point $(N_i, D_i, L_i)$, where the training compute is approximated as $C_i \approx 3N_iD_i$. The black lines visualize the trajectory of our progressive scaling strategy. As either model size or training data volume increases, the validation loss diminishes (left). The derived scaling law enables us to estimate the compute-optimal values of $N$ and $D$ for any given compute budget $C$ (middle and right).

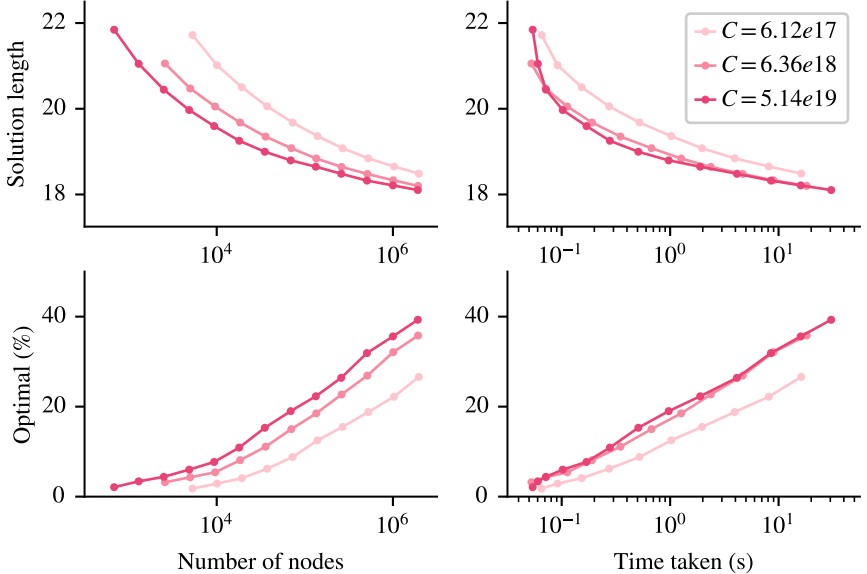

Figure 6: Downstream performance comparison of the three largest models. Each dotted line depicts the mean solution length (top) and the optimality rate (bottom) as a function of node count (left) and time taken (right) for each model. As the training FLOPs and thus the model size increase, our solver needs to expand fewer nodes to reach a specific level of optimality. However, the marginal utility of scaling model size seems diminishing in terms of temporal efficiency.

## 8    Discussion

We have introduced a novel self-supervised learning method to directly solve a combinatorial problem with a predefined goal, reframing the task as *unscrambling*. Training a DNN on mere random scrambles, our method statistically infers a sequence of *reverse moves* that lead back to the predefined goal.

In the Experiment with Rubik's Cube, our method outperformed DeepCubeA, the previous state-of-the-art deep learning method, in terms of both optimality and efficiency, having trained a DNN of the same design with 80% less training data. We also demonstrated the applicability of our method to 15 Puzzle and $7 \times 7$ Lights Out in Additional Evaluations. Our method outperformed DeepCubeA on these problems too, visiting fewer nodes to find optimal solutions. These results provide empirical evidence that our method is generalizable to goal-predefined combinatorial problems.

The Scaling Law analysis indicated that our method is scalable with the number of model parameters and the number of training data samples, as well as the effectiveness of random scrambles as self-supervision signals for the downstream task. However, the benefit of increasing model size on temporal efficiency will plateau at a certain point, depending on the environment.

Nonetheless, our method has some limitations. First, it requires the target problem to have a predefined goal, which might not be the case in combinatorial *optimization* problems like Traveling Salesman Problem whose objective is to find the optimal combination itself. For such problems, classic heuristic or approximation algorithms would remain the best approach (Rego et al., 2011; van Bevern & Slugina, 2020), although there also are deep learning methods leveraging reinforcement learning and graph neural networks (Mazyavkina et al., 2021; Joshi et al., 2022). Second, our method implicitly assumes the reversibility of moves. If the problem's scrambles or solutions do not follow a reversible process, such as when state transitions are non-deterministic and subject to randomness, our method may not be effective.

## 9    Conclusion

We proposed a simple and efficient method for solving Rubik's Cube and other goal-predefined combinatorial problems. Our experiments demonstrated that a DNN can learn to solve such problems only from random scrambles, leveraging the statistical tendency of these scrambles to be optimal. Despite the randomness of training data, our method efficiently generates highly optimal solutions. We hope that our work provides new insights into search and related problems in the field of machine learning.

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
