# OpenReview forum: "Self-Supervision is All You Need for Solving Rubik’s Cube"
_TMLR — Accepted by TMLR_

### Review · Reviewer_43Jt · 2023-05-31

**Summary Of Contributions:**

This paper proposes a method to solve combinatorial problems with a predefined goal, such as Rubik’s Cube. These problems share the commonality that the starting states are obtained by randomly scrambling the _unique known_ solution state, and the task is to recover the solution state using a sequence of operations. The proposed method trains a deep neural network to unscramble – one step at a time.

Main contributions:
* A self-supervised method to collect labeled training data. At inference time, the classifier is used with beam search to iteratively generate candidate paths.
* Experiment results suggest the proposed method is better than the baseline method (DeepCubeA): it uses fewer samples to train the model and the resulting solver is closer to optimal and takes less wall-clock time (after normalization).
* The method shows advantage on two other problems: 15 Puzzle, 7x7 Lights Out.
* Authors also study the scaling law for their models and leverage the law to optimize model size and training data samples given a compute budget.



**Audience:**

Yes

**Claims And Evidence:**

Yes

**Requested Changes:**

* [critical] Please provide analysis to show how much (or little) of the success is due to memorization. At the least, I would like to see a report on how many of the 1K Rubik’s Cube test cases were already seen in the self-supervision stage. Furthermore, it would be great to see how much of the intermediate states during beam-search have been seen before, especially for the intermediate states that are close to the starting states.
* [critical] In Figure 3, there are _far fewer_ red colored dots (proposed method) compared to black and blue ones. (The volumes of the histograms also don't match.) Please clarify.

**Strengths And Weaknesses:**

Strengths:
* The method is simple and the writing is clear to follow.
* The authors discuss important assumptions of the proposed method (section 3). They also give some intuitions on why the method would work given those assumptions.

Weaknesses:
* The experiment design _may be flawed_. It’s not clear if the 1000 Rubik’s Cube test cases are already covered in the self-supervised training set. Although the Rubik’s Cube state space is huge O(10^19), the test cases were obtained by randomly scrambling the solution state with 1000~10000 moves. The states obtained this way are likely not uniform samples from the true state space. Since pseudo random number generators are used, it is entirely possible that the same states (or small perturbations of them) have already been covered in the self-supervision stage. If so, the DNN can just memorize the (reversed) trajectories.
* The proposed self-supervision method probably only works for this special type of combinatorial search problems: unscramble to the know solution state. The neural network may exploit certain biases and structures in the task generation process. I am not convinced the method would work for other type of combinatorial search problems. (In fact, the authors also acknowledged such limitations in section 8).

---

> ### Author Response · Authors · 2023-06-15
> **Rebuttal**
>
> We appreciate your review and constructive feedback on our work. We have carefully considered your concerns and provide the following responses:
>
> > The experiment design *may be flawed*. It’s not clear if the 1000 Rubik’s Cube test cases are already covered in the self-supervised training set. Although the Rubik’s Cube state space is huge O(10^19), the test cases were obtained by randomly scrambling the solution state with 1000~10000 moves. The states obtained this way are likely not uniform samples from the true state space. Since pseudo random number generators are used, it is entirely possible that the same states (or small perturbations of them) have already been covered in the self-supervision stage. If so, the DNN can just memorize the (reversed) trajectories.
>
> > [critical] Please provide analysis to show how much (or little) of the success is due to memorization. At the least, I would like to see a report on how many of the 1K Rubik’s Cube test cases were already seen in the self-supervision stage. Furthermore, it would be great to see how much of the intermediate states during beam-search have been seen before, especially for the intermediate states that are close to the starting states.
>
> We respectfully disagree with your point regarding the potential memorization/leakage of test cases during the training phase.
>
> To begin with, since the diameter of the combinatorial space is 26, random scrambling with a minimum of 1,000 moves—significantly greater than 26—presumably provides a nearly uniform sampling, albeit not perfectly so. To substantiate this claim, we compared the distributions of optimal solution lengths for our test cases and the theoretical estimate [1] using a chi-square test: $\chi^{2}(df=26, N=1000)=17.8$, $p=0.883$. This result confirms that our test cases are indeed representative of the true state space, when insufficient scrambling would raise a significant difference between these.
>
> Accordingly, the probability of a specific test case appearing even once within our training data is minuscule. With 52 billion states(26 states per example $\times$ 2 billion examples) all assumed to be unique for simplicity, the likelihood of such an event is roughly $2.8\times 10^{-26} (\approx\frac{5.2\times 10^{10}}{4.3\times 10^{19}}\frac{1,000}{4.3\times 10^{19}})$. Even when considering a slight non-uniformity in the sampling process, this probability remains negligible.
>
> Addressing the concept of DNNs *memorizing* solutions, we also note that DNNs primarily learn from latent patterns in data, rather than by memorizing specific samples. Even if a test case were to appear in the training set—which we have established as very unlikely—the DNN would not memorize the corresponding solution. At best, it would assimilate the broader patterns inherent in the data, contributing only subtly to overall test performance.
>
> Therefore, we opted not to provide an analysis on memorization.
>
> > In Figure 3, there are *far fewer* red colored dots (proposed method) compared to black and blue ones. (The volumes of the histograms also don't match.) Please clarify.
>
> The red dots are densely clustered and overlapping with each other in a narrow $x$ range, which may give the illusory impression of sparseness.
>
> Regarding the imbalance in the histogram volumes, it is a result of setting different bin widths for the groups. We did so to better represent the distributions with varying densities, each summing up to 1,000 regardless of perceived volumes.
>
> To avoid confusion, we have adjusted the figure to render the histogram more equivalent in volume. After the adjustment, optimal (black) and DeepCubeA (blue) share the same bin width, whereas Ours (red) still has twice narrower bins and thus a smaller perceptual volume.
>
> Similarly, we have also revised Figure 4 (for 15 Puzzle) by increasing the bin width for Optimal (black) to reduce the volume difference from Ours and DeepCubeA. However, to retain visibility, it maintains narrower bins and twice the volume.
>
> > The proposed self-supervision method probably only works for this special type of combinatorial search problems: unscramble to the know solution state. The neural network may exploit certain biases and structures in the task generation process. I am not convinced the method would work for other type of combinatorial search problems. (In fact, the authors also acknowledged such limitations in section 8).
>
> We propose our method specifically for combinatorial problems with a predefined goal and are not claiming its extensibility to other types of problems in this paper.
>
> ---
>
> [1] Tomas Rokicki & Morley Davidson. God's Number is 26 in the Quarter-Turn Metric. https://www.cube20.org/qtm/

---

> > ### Comment · Reviewer_43Jt · 2023-06-15
> > **small sanity check**
> >
> > Thanks for the quick and detailed response.
> >
> > First, thanks for updating Figure 3. It makes sense now.
> >
> > My remaining comments are about memorization and data leakage.
> >
> > I appreciate the probability and statistical analysis, and I get them. However, I don't think they can *fully replace* a sanity check on the actual data.
> >
> > Given trajectories in the form $\ldots, s_t, a_t, s_{t+1}, \ldots$, the proposed method essentially learns $p(a_t | s_{t+1})$. I think it's useful to understand -- in test time -- how many $s_{t+1}$ have been exposed during self-supervision. This helps to measure how much of the success is due to memorization vs generalization.
> >
> > My **minimal request** is still an analysis on the overlap between the 2 billion states collected during self-supervision and the 1K puzzle starting positions. (I am not asking an analysis on the intermediate states during beam search.) I imagine this is not difficult to implement nor computationally expensive to run (e.g. you probably have the data stored, or have a routine to quickly regenerate them). IMHO, if the authors can provide convincing numbers on this, it should strengthen the paper: the neural network learns *generalizable* patterns. Even if there are indeed some overlap, it seems a useful thing to report in the paper and have a discussion about this.

---

> > > ### Author Response · Authors · 2023-06-17
> > > **Sanity check results & Justification**
> > >
> > > We genuinely appreciate your continued engagement and have further considered your concern about potential memorization as below.
> > >
> > > In response to your **minimal request**, we have conducted the sanity check experiment to check for any overlaps between the training data and our test cases. We re-generated 52 billion states (in 2 billion example paths) in the  same way as our main experiment and checked for any intersections with the 1000 test states. As a result, we found 0 overlaps between the randomly generated states and the test set. This result was consistent for 15 Puzzle and 7x7 Lights Out, where we similarly found no test cases within the generated data.
> > >
> > > While we appreciate your suggestion to present these results and discuss potential memorization, we maintain that such a discussion would not enhance the paper's quality. It is improbable that a DNN would *memorize* specific test configurations not just because they are statistically rare but also because high-complexity states are rather unique within our training data. Since memorization by a DNN necessitates repeated exposure to identical inputs, even if some test cases were present in the training data, they are unlikely to be memorized. In the first place, we employ DNNs for efficiently learning generalizable features from a very small subset of the immense state space, as opposed to memorizing individual states.
> > >
> > > We acknowledge, however, that the DNN presumably memorizes some repeated configurations, especially those closest to the predefined goal state. Nevertheless, we posit that such low-complexity states represent the easiest part of the combinatorial search and should not be considered as critical data leakage.

---

> > > > ### Comment · Reviewer_43Jt · 2023-06-17
> > > > **Resolved**
> > > >
> > > > Thank you for conducting the sanity check. My concerns are resolved. (I updated "Claims And Evidence" to Yes)

---

### Review · Reviewer_nQS3 · 2023-06-06

**Summary Of Contributions:**

This paper purposes to solve Rubik's Cube via a beam best-first search using the predicted culminated probability toward the goal state as the evaluation function. Specifically, the novelty of the approach is to train a deep generative model using randomly scrambled trajectories from the goal state, to predict the likely next move to the goal state. Therefore the culminated probability can be evaluated via random rollout using the generative model. The author demonstrated that their approach outperforms the previous state-of-the-art, DeepCube, on the solution length, optimality, number of nodes expanded and the running time. The authors also provide evaluations on other domains. The authors also empirically investigate the scaling law of their method, i.e., the relation between the performance and the model size/training size under a computation budget.


**Audience:**

Yes

**Broader Impact Concerns:**

I do not have ethical concerns about the results of this paper.

**Claims And Evidence:**

Yes

**Requested Changes:**

While I admire the elegant algorithmic approach in this paper, there are a few places that I would like to ask the authors to address:

1. Using randomly scrambled trajectories to train the model doesn't sound very "optimal". I would imagine a another way to produce the quality training trajectories is to use other algorithms (say a standard RL agent) to find an approximate optimal path from the goal state to some arbitrary states. Can the authors elaborate more on why you consider the randomly scrambled trajectories are correlated with the optimal path?

2. It will be great if the author can also include the pseudocode for the beamed best-first search. If I understand correctly, the evaluation function of the search is the culminated probability towards the goal state, whose probability is estimated by the trained DNN? And how do you produce these candidates? Are they generated by random rollouts by the DNN?

3. For the experimental results, are there notions of confidence intervals? E.g., do you compute the average performance by one particular run of the algorithm per instance, or average across multiple runs?

4. I am not sure whether Table 3 makes sense: the number of nodes expanded for your method is 24.26, while for DeepCubeA is 1.14*10^6. Can the author clarify this huge gap.

**Strengths And Weaknesses:**

Strengths: the method is neat and intuitive, and echoes the self-supervision techniques deployed in large language models. The experimental results are impressive.

Weakness: Lack of certain clarity. Few typos/errors in the draft.

---

> ### Author Response · Authors · 2023-06-15
> **Rebuttal**
>
> Thank you for your time and effort spent in reviewing our manuscript. Please find our responses to your comments below.
>
> > 1. Using randomly scrambled trajectories to train the model doesn't sound very "optimal". I would imagine a another way to produce the quality training trajectories is to use other algorithms (say a standard RL agent) to find an approximate optimal path from the goal state to some arbitrary states. Can the authors elaborate more on why you consider the randomly scrambled trajectories are correlated with the optimal path?
>
> We respectfully disagree. Firstly, as we discuss in Section 3, random moves are statistically more likely to be optimal due to shorter paths having a higher probability of connecting to a specific node. Second, generating random paths offers computational advantages in terms of speed and ease over finding optimal or near-optimal paths. Thirdly, the availability of an external solver is not always guaranteed for the target problem at hand.
>
> > 2. It will be great if the author can also include the pseudocode for the beamed best-first search. If I understand correctly, the evaluation function of the search is the culminated probability towards the goal state, whose probability is estimated by the trained DNN? And how do you produce these candidates? Are they generated by random rollouts by the DNN?
>
> We provide a succinct pseudocode of our beam search implementation below:
>
> ```
> procedure BeamSearch(state, DNN, k) {
> 	while not solved:
> 		1. Use DNN to predict probabilities of next moves for each candidate state.
> 		2. Compute cumulative probabilities for the next-depth candidates.
> 		3. Sort and select the top k candidate moves.
> 		4. Apply the selected moves to the corresponding candidates.
> 	return solution;
> }
> ```
>
> The evaluation function is indeed the cumulative product of probabilities of moves, predicted by the DNN. We produce the next-depth candidates by applying the most prospective moves to the corresponding states, rather than through random rollouts.
>
> We do not include a pseudocode like this in the manuscript, as the procedure should be clear from Section 3.
>
> > 3. For the experimental results, are there notions of confidence intervals? E.g., do you compute the average performance by one particular run of the algorithm per instance, or average across multiple runs?
>
> We neither report confidence intervals nor execute multiple runs per test case for our experimental results. This is because the methods we used, including our own, are deterministic, providing consistent results under the same settings. The reported average performances are computed over 1,000 or 500 independent test cases.
>
> > 4. I am not sure whether Table 3 makes sense: the number of nodes expanded for your method is 24.26, while for DeepCubeA is 1.14*10^6. Can the author clarify this huge gap.
>
> Our method indeed expanded significantly fewer nodes than DeepCubeA. This is because our method was able to solve all test cases optimally *without expanding any suboptimal moves*. To make this point more explicit, in our revised manuscript, we include the mean solution length of 24.26 for both Ours and DeepCubeA in Table 3.

---

### Review · Reviewer_v4Mr · 2023-06-15

**Summary Of Contributions:**

The authors show that, surprisingly, you can learn to solve the Rubik's cube and other planning puzzles without any reinforcement learning and just by predicting the action that was taken to get to that state from the solved state. They provide extensive experiments on Rubik's cube, lights out, and 15-puzzle, achieving SoTA results compared to DeepCubeA. They also show scaling laws demonstrating that this method continues to improve with data and size of the network, similar to LLMs. This result is noteworthy because it provides a more scalable approach compared to the deep RL in DeepCube.

**Audience:**

Yes

**Claims And Evidence:**

Yes

**Requested Changes:**

I would like to see more discussion / experiments about the number of scrambles. In the paper the authors set the number of scrambles to be equal to God's number. Intuitively this roughly makes sense, but it would be nice to be more rigorous here. What if there is one weird state with a super high number (so God's number is very high) but every other state is reasonable? It seems that if you set N to be equal to God's number in this case the entropy would be too high to learn anything useful.

This is a very important part of the paper because too low N and too high N simply won't work.

**Strengths And Weaknesses:**

Strengths:
- Surprising results: just predicting the action from random actions is enough.
- Extensive experiments
- Clear writing

Weaknesses:
- The results aren't a lot better than DeepCubeA

---

> ### Author Response · Authors · 2023-06-15
> **Rebuttal**
>
> We are grateful for your thoughtful feedback on our work. Below, we have responded to your comments.
>
> > The results aren't a lot better than DeepCubeA
>
> While we appreciate this viewpoint, we respectfully posit that our results are a lot better than DeepCubeA across all tested problems.
>
> In solving Rubik’s Cube, our method necessitated roughly half the number of nodes to expand to find shorter solutions. For 15 Puzzle, we were able to derive all optimal solutions with a 23% reduction in the number of nodes. When tested on 7$\times$7 Lights Out, our model successfully solved all test cases using a greedy search, where solution length equals the number of nodes, compared to DeepCubeA’s average of $1.14 \times 10^6$ nodes. Importantly, these results were achieved with at most 20% equivalent of the training data used by DeepCubeA.
>
> > I would like to see more discussion / experiments about the number of scrambles. In the paper the authors set the number of scrambles to be equal to God's number. Intuitively this roughly makes sense, but it would be nice to be more rigorous here. What if there is one weird state with a super high number (so God's number is very high) but every other state is reasonable? It seems that if you set N to be equal to God's number in this case the entropy would be too high to learn anything useful.
> >
> > This is a very important part of the paper because too low N and too high N simply won't work.
>
> We concur with your observation that setting N extremely low (or below God's Number) would prevent the DNN from generalizing to higher-complexity states. However, while setting $N$ much higher than God’s Number might compromise training *efficiency,* the test performance would remain largely unaffected due to the diminishing influence of redundant/longer paths.
>
> From the perspective of a specific state/pattern, shorter paths have a higher probability of randomly occurring and thus hold greater weight than longer paths. Assuming there are $M$ moves per state, the probability or relative weight of a $M$-move path happening is $M^x$ times more than of a path consisting of $M+x$ moves. Hence, even if there are a few states with markedly higher complexity, aligning $N$ to these *outlier* states would only minimally impact the performance on lower-complexity states.
>
> Our empirical experiment on 7$\times$7 Lights Out supports this theory.  Despite the high $N$ value set to the upper bound of God's Number—which led to the redundancy of all scramble paths—we still achieved perfect solutions for all test cases.
>
> However, as this may not be obvious enough, we have supplemented the final paragraph of Section 3 (Proposed Method) with the following statements:
>
> > On the other hand, training data covers all possible complexities as long as $N_G\leqslant N$.
> Although redundant paths with more than $N_G$ moves would undermine the training efficiency, they have a negligible impact on the learned distribution due to the significantly lower probabilities than shorter ones.

---

### Decision · Action_Editors · 2023-07-11

**Recommendation:** Accept as is

**Comment:**

I recommend accepting this paper because all the reviewers are positive and both acceptance criteria have been met, but I have some reservations about this paper that I urge the authors to think about.

The result is rather narrow. It applies to needle-in-a-haystack style problems, and specifically it leverages domain knowledge to its full extent: the only thing that makes the domain hard, which is the where the needle is, and how to move out of the haystack. So yes, of course by training a DNN close to the goal states you will be more efficiently solve these problems; classic heuristic search techniques have done exactly that for years (e.g. this is how Checkers was solved). The interesting problem, from the machine learning perspective, is how to do this *generally*, without access to this golden knowledge. Take, as an analogous example, Montezuma's Revenge in the Arcade Learning Environment, widely known to be this kind of problem. If researchers had started from the goal state, this would have been considered cheating, because the purpose of the problem is to figure out how to reach the goal state by interacting with the world and acquiring general knowledge to help get to the goal state.

The paper is incremental in that it improves upon DeepCubeA by being more efficient on this narrow class of problems. It simply removes the distance estimation, but the amount of domain knowledge used is still quite high (just like DeepCubeA).

I'm left unsatisfied -- if I want to solve Rubik's Cube and I'm ok with using privileged information such as starting from the goal state and working backward, why wouldn't I simply use the classic heuristic search methods? I would not have to train a deep network, and the implementation could be specialized to that setting, so it can solve it in a matter of milliseconds, as shown in Table 2.

In the end, the main contribution is that this paper shows a technique to increase the efficiency of neural network work-back-from-the-solution style solvers, and this method was was mostly already proposed by the authors of DeepCubeA.


**Audience:**

This paper will be of interest to a small fraction of the audience of TMLR, specifically those interested in machine learning techniques for single-agent heuristic search problems traditionally solved using variants of A* (Rubik's cube, sliding tile puzzle, etc.)

**Claims And Evidence:**

The authors claim that their method outperforms the current state-of-the-art, DeepCubeA. This is true, in that it is more efficient than DeepCubeA (expanding fewer nodes, taking less wall clock time). The performance itself in terms of quality (solution length and percentage solves) is on par with DeepCubeA.